# DNA methylation and transcriptional trajectories during human development and reprogramming of isogenic pluripotent stem cells

Matthias S. Roost[1], Roderick C. Slieker [2], Monika Bialecka[1], Liesbeth van Iperen[1], Maria M. Gomes Fernandes[1], Nannan He[1], H. Eka D. Suchiman[2], Karoly Szuhai [3], Françoise Carlotti[4], Eelco J.P. de Koning[4,5], Christine L. Mummery[1], Bastiaan T. Heijmans[2] & Susana M. Chuva de Sousa Lopes[1,6]

Determining cell identity and maturation status of differentiated pluripotent stem cells (PSCs) requires knowledge of the transcriptional and epigenetic trajectory of organs during development. Here, we generate a transcriptional and DNA methylation atlas covering 21 organs during human fetal development. Analysis of multiple isogenic organ sets shows that organ-specific DNA methylation patterns are highly dynamic between week 9 (W9) and W22 of gestation. We investigate the impact of reprogramming on organ-specific DNA methylation by generating human induced pluripotent stem cell (hiPSC) lines from six isogenic organs. All isogenic hiPSCs acquire DNA methylation patterns comparable to existing hPSCs. However, hiPSCs derived from fetal brain retain brain-specific DNA methylation marks that seem sufficient to confer higher propensity to differentiate to neural derivatives. This systematic analysis of human fetal organs during development and associated isogenic hiPSC lines provides insights in the role of DNA methylation in lineage commitment and epigenetic reprogramming in humans.

[1] Department of Anatomy and Embryology, Leiden University Medical Center, Einthovenweg 20, 2333 ZC Leiden, The Netherlands. [2] Molecular Epidemiology Section, Leiden University Medical Center, Einthovenweg 20, 2333 ZC Leiden, The Netherlands. [3] Department of Molecular Cell Biology, Leiden University Medical Center, Einthovenweg 20, 2333 ZC Leiden, The Netherlands. [4] Department of Nephrology, Leiden University Medical Center, Albinusdreef 2, 2333 ZA Leiden, The Netherlands. [5] Hubrecht Institute, Uppsalalaan 8, 3584 CT Utrecht, The Netherlands. [6] Department for Reproductive Medicine, Ghent University Hospital, De Pintelaan 185, 9000 Ghent, Belgium. Roderick C. Slieker and Monika Bialecka contributed equally to this work. Correspondence and requests for materials should be addressed to S.M.Cd. S.L. (email: lopes@lumc.nl)

Every organ in the body has a core, organ-specific transcriptional signature that ultimately determines the shape and functionality of each organ and ensures that this remains stable throughout the life of the organism. Whilst much has been published on organ-specific transcriptional and epigenetic landscapes in laboratory animals and stem cell models in vitro, equivalent comprehensive data using a large set of human organs from the same individual (isogenic analysis), that circumvents genetic differences confounding the outcome, has not been performed to date[1–12].

Setting the correct patterns of DNA methylation is crucial during development, but removing those during the reverse process of reprogramming somatic cells from any human tissue to pluripotency as induced pluripotent stem cells (iPSCs)[13, 14] is also important. Reprogramming is accompanied by extensive epigenetic remodeling, which results in a pluripotent state comparable to that of embryonic stem cells (ESCs)[15, 16]. However, although the gene expression signatures of iPSCs and ESCs are similar, when large numbers of lines are compared, individual lines are not necessarily identical[17–20]. This led to the hypothesis that residual epigenetic memory may be retained from the tissue of origin. Indeed, it was demonstrated that mouse and human iPSCs harbor some features of the tissue of origin, i.e. histone modifications, DNA methylation, and microRNAs, which in turn

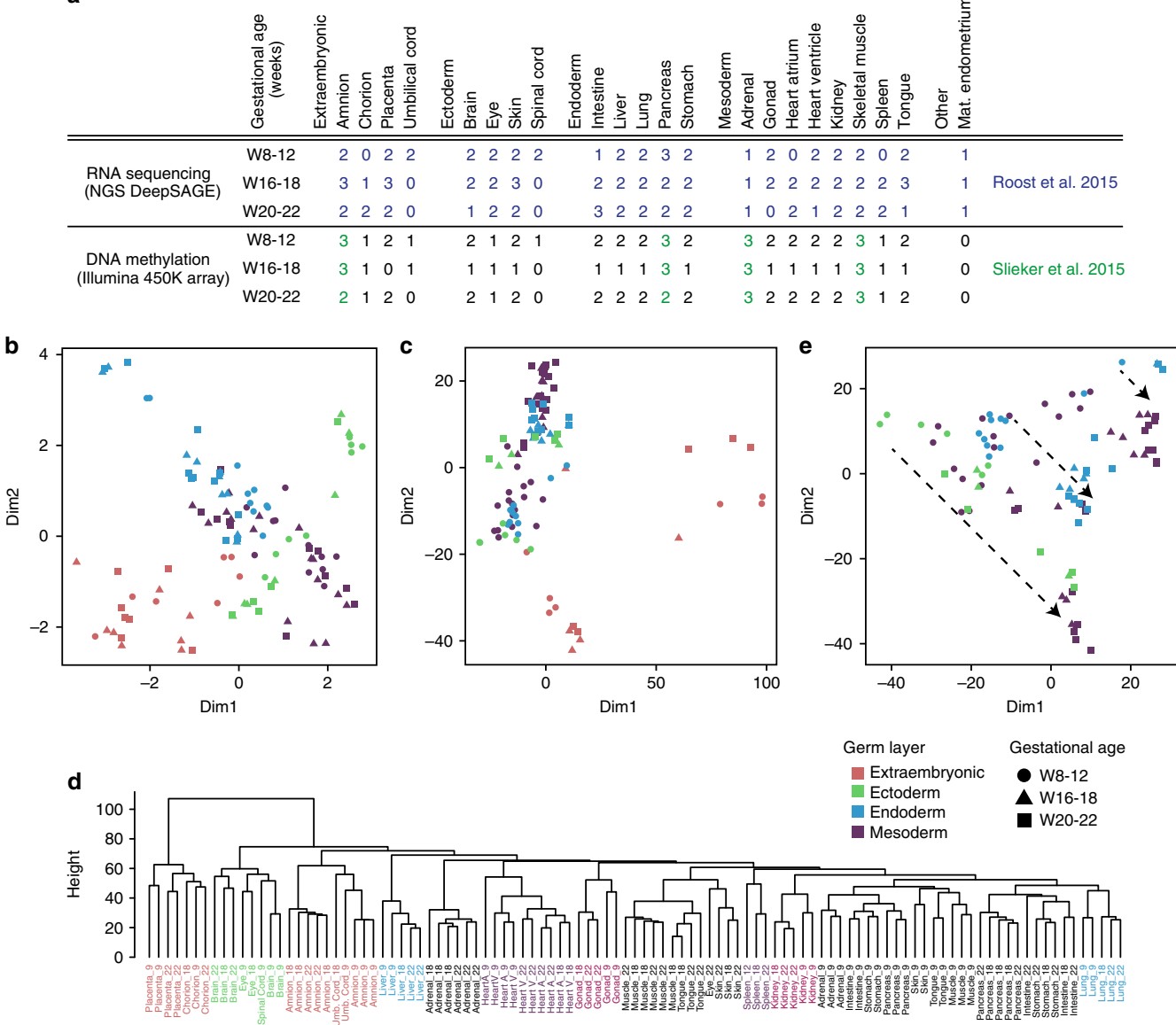

**Fig. 1** Transcriptional and DNA methylation of a collection of organs of first and second trimester. **a** The collection of 111 samples analysed for gene expression (by next generation sequencing (NGS) DeepSAGE) and 105 samples analysed for DNA methylation (by Illumina 450 K array). See also Supplementary Data File 1. Mat, maternal. **b** Multidimensional scaling (Euclidian distance) of the transcriptional profiles of the 111 samples. The colors represent the different germ layers whereas the shapes indicate the gestational age. See also Supplementary Fig. 2a. **c** Multidimensional scaling (Euclidian distance) of the DNA methylation profiles of the 105 samples. The colors represent the different germ layers whereas the shapes indicate the gestational age. See also Supplementary Fig. 2b. **d** Hierarchical clustering (Euclidian distance) of the 105 DNA methylation samples. **e** Multidimensional scaling (Euclidian distance) of the DNA methylation profiles excluding the extraembryonic samples. The colors represent the different germ layers whereas the shapes indicate the gestational age. Dashed arrows represent the trend of DNA methylation dynamics during development in similar organs

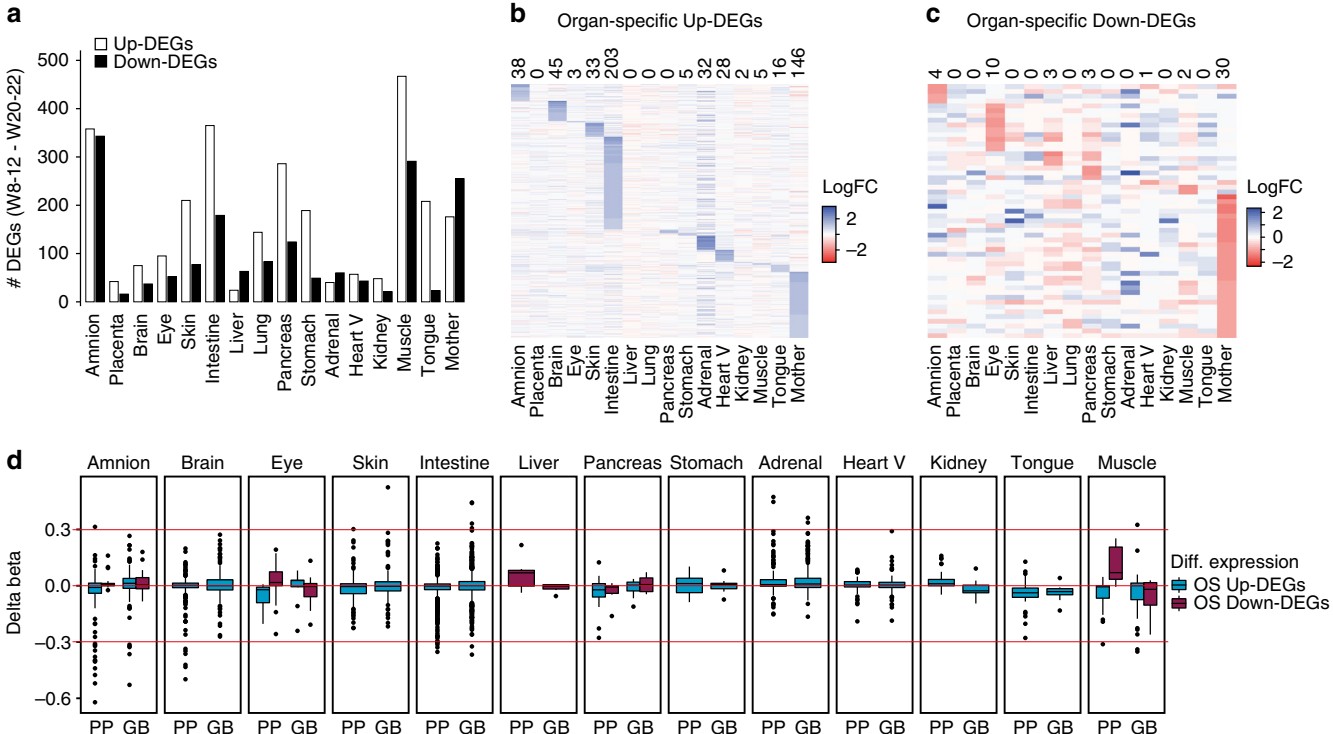

**Fig. 2** Organ-specific upregulated and downregulated genes and associated changes in DNA methylation. **a** Numbers of differentially expressed genes (FDR < 0.01, general linearized model) between W8-12 and W20-22 in different organs. Heart V, heart ventricle; Mother, maternal endometrium. **b** Heatmap of the upregulated genes that were uniquely assigned to one organ. See also Supplementary Data File 2. **c** Heatmap of the downregulated genes that were uniquely assigned to one organ. See also Supplementary Data File 2. **d** Boxplots illustrating the organ-specific methylation changes (delta beta) of the nearest proximal promoter (PP) and the gene body (GB) of the loci identified in **b**, **c**, excluding the maternal endometrium. The red line indicates a delta beta of −0.3, 0 and 0.3. See also Supplementary Data File 3

can favor differentiation towards the lineage from which they were derived[21–29]. The main contribution to the variation between hiPSCs and hESCs has also been suggested to be the genetic background instead of epigenetic memory[16, 30, 31]. However, data to distinguish between these two possibilities is currently lacking. An intriguing difference between mouse and human iPSCs is that the epigenetic memory of mouse iPSCs is lost during continuous passage in culture, whereas human iPSCs appear to have more persistent epigenetic marks[26, 28]. Understanding how these factors influence the differentiation capacity of iPSCs would help determine a better framework for the use iPSCs in disease modeling, drug screening and regenerative medicine.

We have determined the transcriptional profiles of human fetal organs from the first and second trimester of development and identified a set of core organ-specific genes or "key genes" (also referred to as classifier genes) that were highly expressed in the organ it identifies, often from as early as 9 weeks of gestation (W9)[7]. In contrast to the organ-specific transcriptional identity, the core organ-specific pattern of DNA hypomethylation, that remains stable throughout adulthood, takes longer to be established. More precisely, between W9 and W22 the general development-related programs gain DNA methylation and are shutdown, whereas organ-specific genetic programs associated with organ functionality lose DNA methylation[8]. The DNA methylation pattern observed at W22 in some organs appears at least in part to be maintained during adulthood[32–35], suggesting lineage commitment.

Here, we present a comprehensive analysis of human fetal DNA methylation and corresponding genome-wide transcription data: the analysis includes 21 human fetal organs (plus maternal endometrium) from different fetuses (N = 18) at W8-12, W16-18 and W20-22. Furthermore, we used this material to overcome one of the major challenges to assess epigenetic memory in human iPSCs with different origins by generating lines from six isogenic fetal organs (brain, skin, kidney, muscle, lung and pancreas). For all organs, we used the same primary cell isolation method, culture protocol and reprogramming conditions in order to rule out any confounding factors. We compared the DNA methylation profiles of hiPSCs to their organs of origin and showed that hiPSCs derived from fetal brain retain brain-specific DNA methylation marks that led to a higher propensity to differentiate to neural derivatives. This exemplifies how very small differences in DNA methylation may result in different propensity of hiPSCs to differentiate.

## Results

**Human fetal transcriptional and DNA methylation trajectories.** We analyzed DNA methylation (n = 105 samples; Illumina 450 K array) and, if available, the corresponding genome-wide transcriptional data (n = 111 samples; NGS DeepSAGE) from 21 different organs from N = 18 human fetuses at three different time points during gestation W8-12, W16-18 and W20-22 (Fig. 1a and Supplementary Data File 1). 34 of 105 samples of the DNA methylation dataset and the whole transcriptional dataset have been generated previously by us[7, 8] (Fig. 1a).

After quality control (Supplementary Fig. 1), we applied multidimensional scaling (MDS) to the 111 NGS datasets and found that the four directions towards the corners of the MDS plot corresponded to the germ layers (Fig. 1b and Supplementary Fig. 2a). Distinct clusters formed for the liver samples

(endoderm), the brain/spinal cord samples (ectoderm), muscle/heart/tongue samples (mesoderm) and the placenta/chorion/amnion/umbilical cord samples (extraembryonic) towards the periphery of the MDS plot (Fig. 1b and Supplementary Fig. 2a). However, although similar organs often clustered together forming their own spatial domains, many organs intermingled in the center of the MDS plot showing limited separation, in particular organs with mixed origin (pancreas, intestine, lung, adrenal) (Supplementary Fig. 2a).

Next, we performed similar MDS analysis and hierarchical clustering based on Euclidian distance using the sample-matched DNA methylation datasets (Fig. 1c, d). The extraembryonic

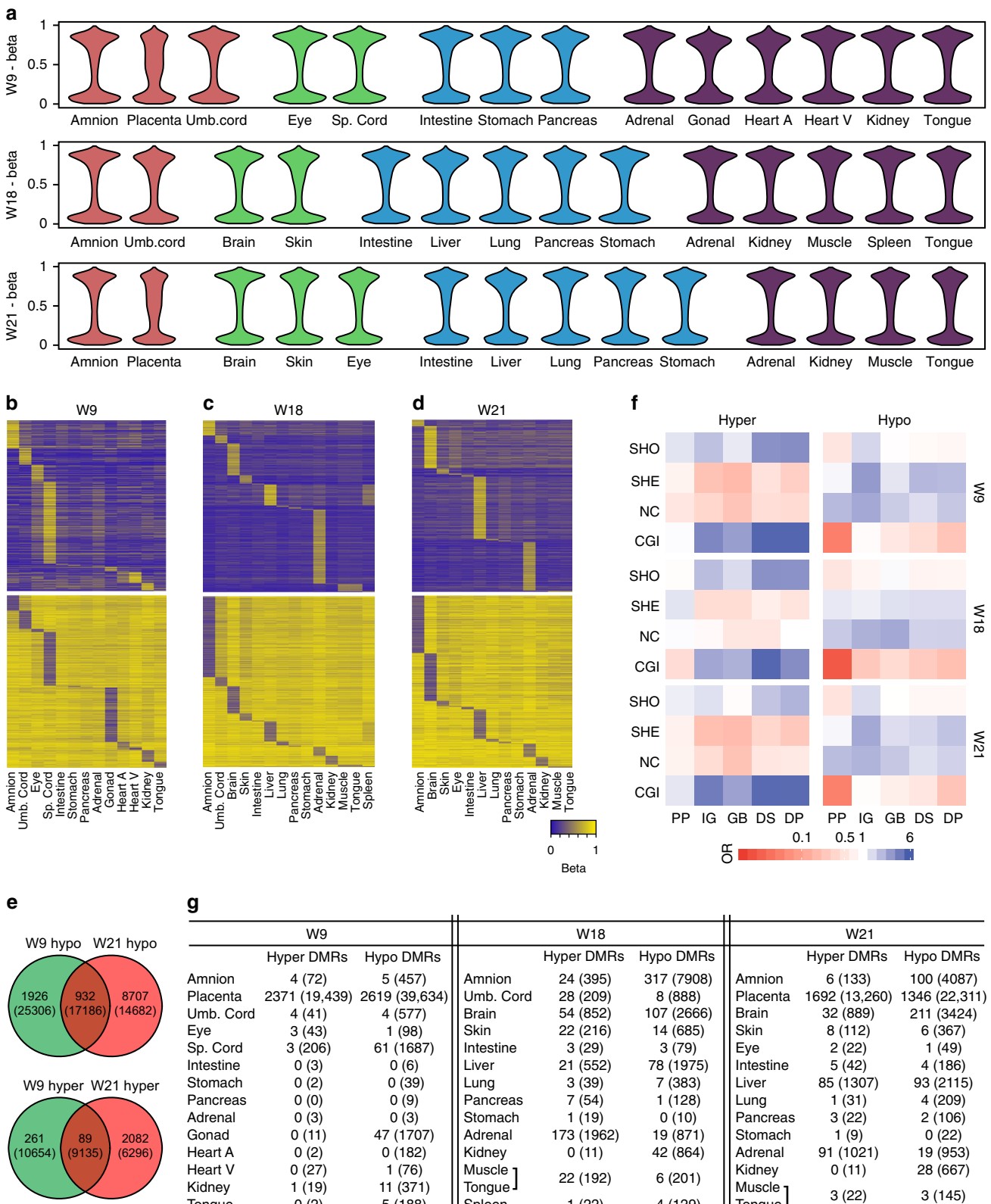

tissues, chorion and placenta, exhibited a fundamentally different DNA methylation pattern from the other (embryonic) organs. They clustered separately (Fig. 1c, d) and showed distinctive high levels of intermediate DNA methylation on autosomes (Supplementary Fig. 1d), previously also described in human term placenta[36]. On the dendrogram, some organs clustered in an organ-specific manner, independently of their developmental age (brain/eye, liver, heart ventricle/atrium, gonad, kidney and lung) and this was similar to the organ-specific clustering observed on the basis of the transcriptome; while other organs including tongue/muscle, stomach/intestine/pancreas, adrenal and skin, clustered at W8-12, separating only at W16-22 in distinct organ-specific clusters. To increase resolution, we then performed a MDS analysis on the DNA methylation dataset excluding the extraembryonic tissues and confirmed the strong dynamics of DNA methylation between W8-22, in contrast to the more stable tissue-specific transcriptome during the same developmental period (Fig. 1e and Supplementary Fig. 2b).

**Small correlation between transcription and DNA methylation.** Next, we investigated the presence of differentially expressed genes or DEGs (upregulated and downregulated) between W8 and W22 per organ (FDR < 0.01; general linearized model) (Fig. 2a). After that, we filtered the DEGs that were organ-specific and had a log-fold change (logFC) > 1 or logFC < −1 (Fig. 2b, c and Supplementary Data File 2). Interestingly, the number of organ-specific DEGs (excluding X-linked genes to avoid sex bias) was small, with the exception of the upregulated (up-)DEGs in the intestine ($n = 203$) and maternal endometrium ($n = 146$). Surprisingly, some organs, like the lung, showed no organ-specific DEGs between W8-22 and many others (liver, pancreas, stomach, kidney and muscle) showed less than 10 organ-specific DEGs, suggesting that either between W8-22 the transcriptional state of the organ-specific progenitors remains similar within each organ or that, alternatively, the bulk sequencing of the organ is masking the cellular maturation of organ-specific progenitors.

We examined whether the changes in expression of organ-specific DEGs were accompanied by changes in DNA methylation in their respective loci. This was done by plotting the calculated differences in beta values between W8-12 and W20-22 (delta beta) of the CpGs in the proximal promoters (PP) (−1.5 Kb to + 0.5Kb) and gene bodies (GB) (+ 0.5Kb to 3′ untranslated region (UTR)) of the identified organ-specific up-DEGs and down-DEGs (Fig. 2d and Supplementary Data File 3). Overall, a direct correlation between organ-specific transcriptional up- and downregulation and changes in DNA methylation was in most organs not existent or very modest. However, the reduced number of organ-specific down-DEGs was still sufficient to be associated with an increase in DNA methylation (positive delta beta) in their PP and a decrease (negative delta beta) in their GB in eye, liver and muscle (Fig. 2d). By contrast, organ-specific up-DEGs were accompanied by a modest decrease of DNA methylation (negative delta beta) in

their PP in amnion, eye, pancreas, tongue and muscle; but an increase (positive delta beta) in their GB only in amnion, eye and pancreas (Fig. 2d).

Exemplifying the association between gene expression and DNA methylation, two CpGs in the PP of the muscle up-DEG *C8orf2*, also shown by others to be overexpressed in skeletal muscle[37], showed reduced levels of DNA demethylation between W8-12 and W20-22 (−0.31 and −0.19 of delta beta). In the brain, up-DEG *GFAP*, important in the development of the central nervous system[38, 39] and up-DEG *CHRM1*, important in schizophrenia[40], showed pronounced demethylation in their PP between W8-12 and W20-22. The PP of *GFAP* contained five CpGs that underwent demethylation (−0.50, −0.43, −0.42, −0.36, −0.35 of delta beta) and the PP of *CHRM1* contained four CpGs that were demethylated (−0.34, −0.26, −0.24, −0.16 of delta beta). In the eye, we observed increased methylation in five CpGs in the PP ( + 0.19, + 0.17, + 0.17, + 0.13 and + 0.11 of delta beta) and increased demethylation in two CpGs in the GB (−0.2 and −0.13 of delta beta) of eye down-DEGs *CRYBB3* and *CRYBA1*, which are structural components of crystalline, between W8-12 and W20-22.

**Isogenic human organs have specific DNA methylation patterns.** One of the unique features of our datasets is the isogenic nature of many sets of organs (Supplementary Data File 1) and in particular three sets of DNA methylation of 14 organs from a W9 male, W18 female and W21 male (Fig. 3a). This enabled us to compare the methylation status of individual CpG between isogenic organs with the assurance that any difference in DNA methylation would be entirely attributable to differences between the organs (Fig. 3a and Supplementary Fig. 3a).

We identified hypermethylated and hypomethylated CpGs, defined as a pairwise (organ of interest versus all others of the isogenic set) difference in beta values of > 0.2 or < 0.2, respectively, a cut-off that we have shown previously to be sufficient to highlight differences between different organs[8] (Fig. 3b–d and Supplementary Fig. 3b,c). The placenta was part of the isogenic W9 and W21 sets and showed high numbers of hypermethylated and hypomethylated CpGs (W9: 19439 and 39634 CpGs, W21: 13260 and 22311 CpGs, respectively) compared to the other organs (Supplementary Fig. 3b,c), most likely reflecting its extraembryonic origin. Therefore, we generated separate plots excluding the hypermethylated and hypomethylated CpGs of the placenta (Fig. 3b, d).

The number of organ-specific hypomethylated CpGs was consistently higher than the number of hypermethylated CpGs (Supplementary Fig. 3d), highlighting the importance of hypomethylation as a distinguishing feature between organs[5, 8, 41]. In most individual organs (excluding the placenta), the number of organ-specific hypermethylated and hypomethylated CpGs increased between W9 and W21 (Fig. 3e and Supplementary Fig. 3d), suggesting progression in organogenesis and lineage commitment.

**Fig. 3** DNA methylation signatures of isogenic sets of organs. **a** Violin plots showing the distribution of the DNA methylation (beta values) in each of the 14 isogenic organs at W9, W18 and W21. Heart A, heart atrium; Heart V, heart ventricle; Sp. cord, spinal cord. **b–d** Heatmaps illustrating hyper- (top) and hypomethylated (bottom) CpGs per organ with a beta value difference of > 0.2 or < 0.2, respectively, at W9 **b**, W18 **c** and W21 **d**. Due to the quantity of hypermethylated and hypomethylated CpGs in the placenta at W9 and W18, those were removed but are given in Supplementary Fig. 3b, c. **e** Venn diagram illustrating the overlap of the hypermethylated (bottom) and hypomethylated (top) CpGs of the organs at W9 and W21. The numbers in brackets represent the CpGs including the placenta. The overlapping CpGs per organ are given in Supplementary Fig. 3d. **f** Genic and CGI-centric annotation of the identified hyper- (left) and hypomethylated (right) CpGs given as odds ratio (OR). CGI, CpG island; DP, distal promoter; GB, gene body; IG, intergenic; NC, non-CGI; PP, proximal promoter; SHE, shelf; SHO, shore. **g** Number of organ-specific hypermethylated and hypomethylated differentially methylated regions (DMRs) per isogenic organ. Tongue and muscle were pooled. For the nearest associated loci see Supplementary Data File 4. The number of organ-specific hypermethylated and hypomethylated individual CpGs are given in brackets

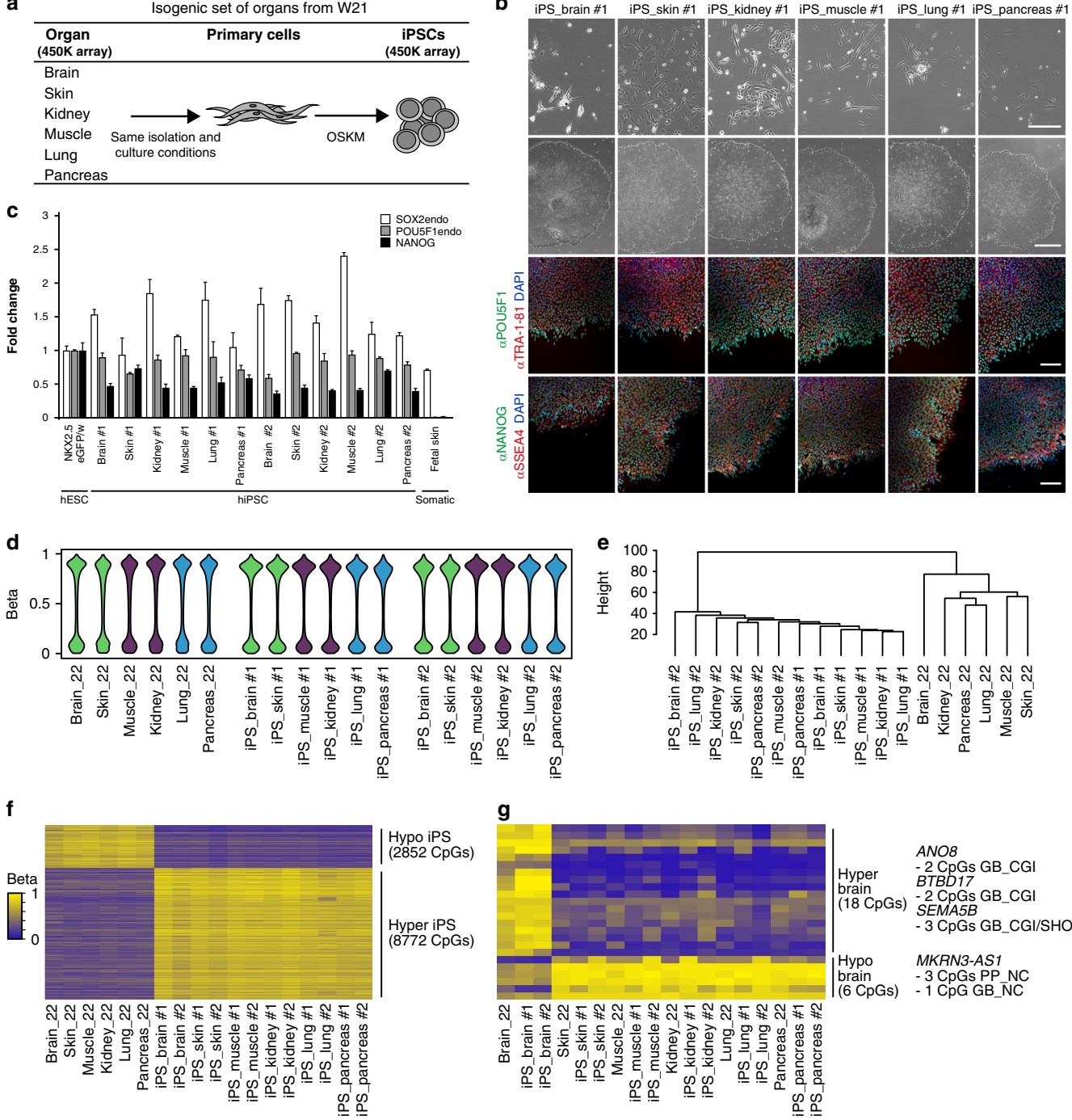

**Fig. 4** Epigenetic memory of isogenic human iPSCs from six organs **a** Illustration describing the experimental setup for reprogramming. OSKM, reprogramming factors POU5F1, SOX2, KLF4, MYC. **b** Bright-field pictures of the primary cells (first row, scale bar 50 μm) and the iPSC colonies of clone #1 (second row, scale bar 1 mm) from the six organs. The iPSCs (clones #1) were immunostained for POU5F1 and TRA-1-81 (third row, scale bar 100 μm) and NANOG and SSEA4 (fourth row, scale bar 100 μm). For the single channels and the immunostaining of clones #2 see Supplementary Fig. 4a. **c** Real-time quantitative PCR analysis of the 12 iPSCs for endogenous *SOX2*, endogenous *OCT4* and *NANOG*. NKX2.5^eGFP/w hESCs were used as positive and fetal skin as negative control. Each bar represents the mean of three technical replicates ± standard deviation. **d** Violin plots depicting the distribution of the DNA methylation (beta values) of the 12 iPSCs and their isogenic organ of origin. **e** Hierarchical clustering of the 12 iPSCs and their isogenic organ of origin based on the DNA methylation profiles. **f** Heatmap of uniquely hypermethylated (top) and hypomethylated (bottom) CpGs in the 12 iPSCs compared to the six organs of origin with a beta value difference of > 0.2 or < 0.2, respectively. See also Supplementary Data File 5. **g** Heatmap depicting uniquely hypermethylated (top) and hypomethylated (bottom) CpGs in the brain triplet (two iPSC clones and brain) compared to the five triplets with a beta value difference of > 0.2 or < 0.2, respectively. The genic location and gene identity are provided

We investigated the genomic location of the organ-specific hypermethylated and hypomethylated CpGs identified (including the placenta) per developmental stage. The organ-specific hypermethylated CpGs were mainly associated with CpG islands (CGIs) and their shores, but only downstream regions ($p < 0.5 \times 10^{-10}$; chi-squared test) showed significant enrichment at all three time points. By contrast, the organ-specific hypomethylated CpGs were enriched in non-CGI regions and CGI-shelves consistently at all three time points, particularly when they are associated with distal promoters ($p < 0.5 \times 10^{-10}$; chi-squared test) and downstream regions ($p < 0.005$; chi-squared test) (Fig. 3f). Using the chromatin state segmentations for the available organs from the Epigenomics Roadmap Project[42], the identified hypomethylated CpGs were enriched at enhancers confirming previous studies with less organs[8] (Supplementary Fig. 3e). Hypermethylation on the other hand was rather associated with repressed Polycomb regions and bivalent TSS and bivalent enhancers (Supplementary Fig. 3e).

In addition to DNA methylation changes in single CpGs in the isogenic datasets, we also inferred the organ-specific differentially methylated regions (DMRs), defined as three consecutive hypermethylated or hypomethylated CpGs within 1Kb of each other, applying a widely-used algorithm[35] (Fig. 3g). Considering the limited number of organ-specific hypermethylated and hypomethylated CpGs at W9, many W9 organs contained no organ-specific DMRs (Fig. 3g). However, the number of organ-specific DMRs increased between W9 and W21, and in the second trimester each organ showed at least one organ-specific DMR (Fig. 3g). The skeletal muscle and tongue, both muscular organs, showed similar DNA methylation patterns and were therefore pooled. The organ-specific DMRs were mapped to their nearest loci (Supplementary Data File 4) and, interestingly, many identified loci showed biological relevance for the corresponding organ. At W21, one of the intestine hypomethylated DMRs was mapped to *TFF1*, a gene that has been reported to be downregulated via DNA methylation in gastric cancer[43]. The stomach at W21 on the other hand harbored a hypermethylated DMR associated with *NKX3-2*, which is thought to be involved in stomach development[44].

**Isogenic set of human iPSCs from six different W21 organs.** After determining the trajectory of DNA methylation during (isogenic) lineage commitment in development, we investigated the direct effects of reprogramming cells from different isogenic organs to iPSCs with respect to DNA methylation. We developed a protocol to isolate and culture primary cells from different human fetal organs at W21 and reprogrammed primary cells from six different organs representing the three primary embryonic germ layers: brain and skin (ectoderm), kidney and muscle (mesoderm), and lung and pancreas (endoderm) (Fig. 4a, b). After lentiviral reprogramming using *POU5F1* (or *OCT4*), *KLF4*, *SOX2* and *c-MYC*[45], two clones per iPSC line were expanded and characterized (Fig. 4b, top-middle row). Immunohistochemical analysis showed that all clones expressed the pluripotency-associated markers POU5F1, TRA-1-81, NANOG and SSEA4 (Fig. 4b bottom rows and Supplementary Fig. 4a). Furthermore, (endogenous) *POU5F1*, *NANOG* and *SOX2* were also detected at the transcriptional level using real-time quantitative PCR (Fig. 4c). From the 12 hiPSC clones generated, 11 clones showed normal karyotype (46,XY) and no major genomic aberrations (Supplementary Fig. 4b). Clone #2 derived from the lung acquired an unbalanced translocation involving chromosome X and 16, der(16)t(X;16), in all cells examined (25 metaphases) (Supplementary Fig. 4b).

**Epigenetic memory in human iPSCs generated from brain tissue.** To determine whether differences in residual DNA methylation were evident between each isogenic organ and its derivative hiPSC clones (epigenetic memory), the DNA methylation profiles obtained for each of the hiPSC lines generated (two different clones per line) were compared with the organ of origin. The total DNA methylation distribution showed that the hiPSCs were more hypermethylated than their organ of origin (Fig. 4d). Hierarchical clustering using the DNA methylation datasets revealed a separation between the 12 hiPSC clones and their isogenic tissue counterparts (Fig. 4e). We included two previously published 450 K datasets of human pluripotent stem cells (in total $n = 39$ hiPSCs and $n = 32$ hESCs)[5, 46] in the comparison and observed that the different hESCs and hiPSCs lines clustered together (Supplementary Fig. 4d) indicating that general patterns of DNA methylation of all lines studied might be comparable.

To explore this further, we looked for a hiPSC-specific DNA methylation barcode by comparing the CpG methylation levels of our hiPSC clones to that of the six tissues of origin (Fig. 4f). We found 8772 hypermethylated ($> 0.2$ delta beta) and 2952 hypomethylated ($< 0.2$ delta beta) CpGs common to all 12 iPSC clones (Supplementary Data File 5), underlining the role of DNA methylation during reprogramming towards pluripotency[47]. The hypermethylated and hypomethylated CpGs identified in our hiPSC clones showed comparable methylation levels in the large majority of the samples from two external datasets analyzed[5, 46] (Supplementary Fig. 4d), suggesting that the DNA methylation pattern of hiPSC-specific CpGs could barcode other human PSCs. Gene ontology analysis revealed that hypermethylated CpGs were enriched in genes regulating cell adhesion, whereas hypomethylated CpGs were enriched in genes regulating embryonic morphogenesis and development (Supplementary Data File 6).

Next, we focused on the CpGs that were hypermethylated ($> 0.2$ delta beta) or hypomethylated ($< 0.2$ delta beta) in each triplet (organ of origin and the two associated hiPSC clones) compared to the other five triplets, which would reflect organ-specific epigenetic memory. Surprisingly, only the brain-triplet contained distinctive methylated CpGs: 18 hypermethylated and 6 hypomethylated CpGs (Fig. 4g). However, 7 of the 18 brain-triplet hypermethylated CpGs were mapped to the gene bodies of *BTBD17*, a protein-coding gene overexpressed in fetal and adult brain[37, 48]; *ANO8*, a $Ca^{2+}$-activated chloride channel expressed in the human brain[37, 49]; and *SEMA5B*, a semaphorin associated with axon development[50]; and 3 of the 6 brain-triplet hypomethylated CpGs mapped to the proximal promoter of *MKRN3-AS1*, an antisense RNA that is overexpressed in brain and spinal cord[37]. Those genes as well as the genic location of the CpGs, i.e. gene bodies for hypermethylation and proximal promoters for hypomethylation, suggested that there is a small degree of brain-specific residual epigenetic memory in the brain-hiPSCs.

To test the functionality of the brain-specific residual epigenetic memory in the brain-hiPSCs, we differentiated both clones of the brain-hiPSCs and skin-hiPSCs to the neural lineage (Supplementary Fig. 4e). After 7 days of differentiation, the brain- and skin-hiPSCs generated similar percentages of embryoid bodies composed of at least 50% of neural rosettes (Supplementary Fig. 4f). However, the brain-hiPSC culture showed higher *TUBB3* expression and less *SOX9* expression than the skin-hiPSC culture, suggesting higher neural differentiation propensity (Supplementary Fig. 4g). Indeed, after 12 days of differentiation, the brain-hiPSC culture showed more *TUBB3* expression (Supplementary Fig. 4g) and contained more GFAP-positive neural derivatives (Supplementary Fig. 4h) than the skin-hiPSC culture. In summary, both the brain-hiPSCs and

skin-hiPSCs differentiate to neural derivatives, but the brain-hiPSCs did so with slightly higher propensity.

## Discussion

In this study, we generated a human atlas consisting of large number of DNA methylation ($n = 105$ samples) and transcriptome ($n = 111$ samples) profiles of human fetal organs from 18 individuals, at matched gestational time points (W8-12, W16-18, W20-22).

The analyses of the transcriptional datasets allowed us to identify DEGs that were upregulated or downregulated in an organ- and developmental stage-specific manner. We further compared these differences in gene expression with changes in DNA methylation per organ analysed. Organ-specific down-DEGs, in particular in the eye, muscle and liver, associated with increased DNA methylation in their proximal promoters and decreased DNA methylation in their gene bodies, in line with previous findings using hESCs, fibroblasts and monocytes[51]. Organ-specific up-DEGs were accompanied by decreased DNA methylation in their proximal promoters and increased DNA methylation in their gene bodies in amnion, eye and pancreas. This modest correlation between up-DEGs and DNA methylation changes may be partly explained by the fact that many CpGs in proximal promoters are unmethylated, independent of transcriptional state of each loci.

Our isogenic sets of 14 organs from W9, W18 and W21 provided us with the organ-specific DNA methylation pattern and revealed its dynamics between W9 and W21. Future studies using further improvements in technology to investigate genome-wide DNA methylation and transcription simultaneously at the single-cell level will increase the resolution from organ to specific progenitor cell types of interest[52, 53].

We succeeded in deriving hiPSCs from six isogenic organs (brain, skin, kidney, muscle, lung and pancreas) from W21, a gestational time point with pronounced organ-specific differences in both hypermethylation and hypomethylation of DMRs. Our results suggest that the 12 hiPSC clones derived were comparable with respect to global DNA methylation to other human PSCs and different from the organs of origin.

Taking advantage of the experimental setup with a large number of organs using the same primary cell isolation method, culture protocol and reprogramming conditions, we finally investigated whether DNA methylation would relate isogenic derivative hiPSC clones to their organs of origin (epigenetic memory). We observed that the brain-hiPSC clones retained 18 hypermethylated and 6 hypomethylated CpGs that were brain-specific. This could be an underestimate due to the nature of the platform used. Notably, the nearest loci of those CpGs did not appear to be random and 11 of them were associated with four genes (*ANO8, BTBD17, SEMA5B, MKRN3-AS1*) having a neural context. In addition, the fact that the other organ-derived hiPSCs did not show residual DNA marks relating them to their organ of origin, but rather resembled skin-hiPSCs may suggest that the reprogrammed cells may have been fibroblasts or stromal cells independent of the organ of origin. In future studies, we would consider reprogramming either more homogeneous somatic cells or defined cell types. Nevertheless, when subjected to the same directed differentiation protocol, the brain-hiPSCs and skin-hiPSCs showed different propensity to form neural derivatives. This difference in differentiation propensity, perhaps caused by the limited epigenetic memory of the brain-hiPSCs, suggested that this phenomenon should be more extensively studied as it may have implications in the use of human iPSCs in biomedical applications[54, 55]: the tissue of choice from which to derive patient-specific hiPSCs and the retention of specific DNA

methylation marks may determine the success of each specific application (type of drug-testing, type of disease modeling or tissue repair). Furthermore, it would be interesting to reprogram non-isogenic cells from the same tissue in order to investigate to what extent the genetic component compared to the epigenetic one has implications on the differentiation propensity of iPSCs.

## Methods

**Ethical statement**. This study has been approved by the Medical Ethical Committee of the Leiden University Medical Center (P08.087). Informed consent was obtained from all patients and was compliant with the Declaration of Helsinki (World Medical Association).

**Fetal tissue procurement and primary cell culture**. Human fetal organs used for DNA extraction were processed as follows: after washing with 0.9% NaCl (Fresenius Kabi, France), the identified organs were immediately snap-frozen and stored at −80 °C[8]. For primary cell culture, pieces of human fetal organs were minced with a scalpel (Swann Morton, Sheffield, UK) and each transferred to gelatin-coated wells of 6-well plates with isolation medium (Dulbecco's Modified Eagle Media (DMEM)/F12 supplemented with Glutamax (Gibco, Bleiswijk, the Netherlands), 10 mM NEAA, 2 mM L-glutamine, 25 U per ml penicillin, 25 mg per ml streptomycin, 50 µg per ml gentamicin, 100 mM b-mercaptoethanol (all Invitrogen, Breda, The Netherlands), 25 µg per ml normocin (Invivogen, San Diego, USA), 20% knock-out serum replacement (KOSR; Invitrogen, Breda, the Netherlands)). After two days, the medium was changed and one day later, the cells were washed with Dulbecco's phosphate buffered saline (DPBS, Gibco) and fresh isolation medium was added. After six to seven days, the cells were trypsinized using 0.25% Trypsin-EDTA (Gibco) and frozen in freezing medium (80% isolation medium, 10% KOSR, 10% DMSO (Sigma-Aldrich, St. Louis, USA)).

**Generation of hiPSCs**. To reprogram the fetal cells, the pRRL.PPT.SF.hOKS-MidTomato-preFRT polycistronic lentiviral vector was used[45]. Briefly, $2 \times 10^4$ cells per 12-well plate well were plated and transduced the following day with the lentivirus at 1-2 MOI in isolation medium supplemented with 4 µg per ml polybrene (Sigma-Aldrich, St. Louis, USA). After 24 h, the medium was changed and three days later, all the cells were split 1:1 or 1:2 (depending on the fluorescence intensity) into 60 mm dishes coated with mouse embryonic fibroblasts (MEFs; $7.2 \times 10^5$ MEFs per dish). After culturing the transduced cells in isolation medium for one day, the medium was changed to hiPSC medium (Dulbecco's Modified Eagle Media (DMEM)/F12 supplemented with Glutamax, 10 mM NEAA, 25 U per ml penicillin, 25 mg per ml streptomycin, 100 µM b-mercaptoethanol, 20% knock-out serum replacement and 10 ng/ml basic FGF (PreproTech Neuilly-Sur-Seine, France)). After manual picking, hiPSC-like colonies were cultured for one to three passages and then either frozen in 90% fetal calf serum (Sigma-Aldrich) and 10% DMSO (Sigma-Aldrich) or further expanded on Matrigel (Corning, Wiesbaden, Germany) in mTESR1 (Stem Cell Technologies, Grenoble, France).

**Characterization and neural induction of hiPSCs**. Immunocytochemistry was performed following standard procedures. Briefly, cells were fixed in 4% (w/v) paraformaldehyde (PFA, MERCK, Darmstadt, Germany) for 15 min at room temperature (RT). Subsequently, the cells were first permeabilized using 0.1% Triton X-100 (Sigma-Aldrich, St. Louis, USA) and then blocked with 1% bovine serum albumin, fraction V (BSA, Sigma-Aldrich) in 0.05% Tween-20 (Promega, Madison, USA) for 1 h. The following primary antibodies were then applied overnight at 4 °C: Goat anti-OCT4 (Stock 0.1 mg per ml, 1:100, SC-8628, Santa Cruz Biotechnologies, Dallas, USA), mouse anti-TRA-1-81 (Stock 1 mg per ml, 1:100, MAB4381, Millipore, Bedford, USA), mouse anti-SSEA4 (Stock 0.2 mg per ml, 1:100, SC-59368, Santa Cruz Biotechnologies), rabbit anti-NANOG (Stock 0.5 mg per ml, 1:250, 09-0020, Stemgent, San Diego, USA). The secondary antibodies Alexa Fluor 488 donkey anti-rabbit (Stock 2 mg per ml, 1:500, A-21206, Life Technologies, Carlsbad, USA), Alexa Fluor 488 donkey anti-goat (Stock 2 mg per ml, 1:500, A-11055, Life Technologies), and Alexa Fluor 594 donkey anti-mouse (Stock 2 mg per ml, 1:500, A-21203, Life Technologies) were added for 1 hour at RT and the nuclei were counterstained using 4′,6-diamidino-2-phenylindole (DAPI, Life Technologies). Imaging was performed on an Eclipse Ti imaging system (Nikon, Tokyo, Japan) operated by NIS Elements software and compiled in Photoshop CS6 (Adobe Systems Inc., San Jose, USA).

RNA from the hiPSCs was extracted with the RNeasy Kit (Qiagen, Hilden, Germany) including on-column DNase digestion, followed by cDNA generation with the iScript™ cDNA Synthesis Kit (BioRad, Hercules, USA). Quantitative PCR was carried out on the CFX96TM Realtime system, C1000TM Thermal Cycler (Biorad) using the iQ SYBR Green Supermix (BioRad) and the following program: (1) 3 min. 95 °C; (2). 40 cycles 15 s 95 °C; 30 s 60°C, 45 s 72 °C; and (3) 10 s 95 °C; 5 s 65 °C, 50 s 95 °C). The ΔΔCt method and normalization to *GAPDH* and *ACTB* was used to assess expression levels. The expression levels of hiPSCs were compared to those of the hESC-NKX2.5eGFP/w line, which was used as positive control[56]. The primer sequences are: *ACTB* - Fw CTG GAA CGG TGA AGG TGA

CA and Rv AAG GGA CTT CCT GTA ACA ACG CA; *GAPDH* - Fw CTG CAC CAC CAA CTG CTT AG and Rv GTC TTC TGG GTG GCA GTG AT; *POU5F1* endogenous – Fw GAC AGG GGG AGG GGA GGA GCT AGG and Rv CTT CCC TCC AAC CAG TTG CCC CAA AC; *NANOG* - Fw TGC AAG AAC TCT CCA ACA TCC T and Rv ATT GCT ATT CTT CGG CCA GTT; *SOX2* endogenous – Fw GGG AAA TGG GAG GGG TGC AAA AGA GG and Rv TTG CGT GAG TGT GGA TGG GAT TGG TG[13, 57, 58].

The karyotypes of the 12 hiPSC lines were assessed using combined binary ratio labelling (COBRA)[59]. Briefly, glass slides containing air-dried metaphase spreads were incubated with 100 μg per ml RNase I (Roche, Woerden, the Netherlands) in 2 x Saline-Sodium Citrate (SSC, Sigma-Aldrich, St Louis, USA) at 37 °C for 10 min, followed by incubation with 0.005% pepsin (Sigma-Aldrich) in 0.1 M HCl for 5 min at 37 °C and treatment with 1% formaldehyde (Sigma-Aldrich) in PBS at RT for 10 min. After hybridization with the chromosome-painting probe pools for 40–72 h, slides were washed in 2 x SSC; and 0.1% Tween-20 (Promega, Madison, USA) in 2 x SSC. The chromosomes were visualised using a Leica DMRA fluorescence microscope (Leica, Wetzlar, Germany) with the CoolSnap HQ2 camera (Photometrics, Tucson, USA), and the COBRA-FISH software (Applied Imaging, San Jose, CA).

Neural induction of hiPSCs was performed using the Stemdiff Neural System (Stemcell Technologies, Catalog #05835, Vancouver, Canada) according to manufacturer's instructions. Briefly, embryoid bodies (4000 cells per embryoid body) were grown for 4 days and subsequently plated on matrigel-coated dishes. Cells were analysed by dark-field microscopy and scored for the presence of neural rosettes at day 7 (% EBs composed of at least 50% rosettes, N = 3). After 7 and 12 days of neural differentiation, cells were fixed with 4% PFA (20 min, RT) and used for immunofluorescence as above, using as primary antibody rabbit anti-GFAP (Stock 2.9 mg per ml, 1:200, Z0334, DAKO, Heverlee, Belgium), rabbit anti-SOX9 (Stock 1 mg per ml, 1:200, AB5535, Millipore, Bedford, USA) and mouse anti-TUBB3 (Stock 1 mg per ml, 1:200, AB78078, Abcam, Cambridge, UK) and as secondary antibody Alexa Fluor 488 donkey anti-rabbit (Stock 2 mg per ml, 1:500, A-21206, Life Technologies, Carlsbad, USA) and Alexa Fluor 594 donkey anti-mouse (Stock 2 mg per ml, 1:500, A-21203, Life Technologies). Quantitative PCR for neural markers was performed as described above and the data were presented as mean ± standard deviation of technical triplicates. Statistical analysis was performed by a Student's t-test (two-tailed) using the statistical software package SPSS 20.0 (SPSS Inc., Chicago, IL, USA). p < 0.05 was considered significant. *TUBB3*–Fw GGC CAA GGG TCA CTA CAC G and Rv GCA GTC GCA GTT TTC ACA CTC; *SOX9*–Fw AGC GAA CGC ACA TCA AGA C and Rv-CTG TAG GCG ATC TGT TGG GG.

**DNA extraction and 450 K array data processing**. The genomic DNA (gDNA) of the different organs was extracted as previously described[8]. Briefly, after homogenization, lysis with proteinase K (600 mAU per ml, Qiagen, Hilden, Germany), and degradation of residual RNA using RNase A (10 mg per μl, Invitrogen, Carlsbad, USA), gDNA was extracted on phenol/chloroform basis with Phase Lock Heavy Gel 2 ml Eppendorf tubes (5PRIME, Hilden, Germany). For the hiPSCs (from passage 10-14), the Wizard Genomic DNA Purification Kit (Promega, Leiden, the Netherlands) was used following manufacturer's instructions[60]. In addition to the Nuclei Lysis Solution of the kit, proteinase K (Qiagen) was added to lyse the cells followed by removing the remaining RNA with RNase A. The gDNA was quantified using the Qubit dsDNA BR Assay Kit on a Qubit 2.0 Fluorometer (Sigma-Aldrich, St Louis, USA). An average input of 600 ng was used for bisulfite conversion with the EZ-96 DNA methylation kit (Zymo Research, Orange County, USA). Subsequently, the DNA methylation profiles were determined with the Illumina HumanMethylation450 BeadChip according to the manufacturer's protocol.

The package minfi[61] was used to import the data in R version 3.2.2. To normalize the data, a custom pipeline was used[8, 62, 63]. Briefly, after filtering probes with a low bead count (< 3), high detection p-value (> 0.01), and with a low success rate (< 95%), and ambiguously mapped probes, background correction and color correction were applied. Eventually, the data was normalized using functional normalization as implemented in the minfi package[61]. For all the analyses, except for the DNA methylation profiles of isogenic samples, the probes in CG SNPs (with an allele frequency > 5%) as well as CG probes on the sex chromosomes were excluded[64]. The DNA methylation data of the fetal organs (n = 105 samples) and the stem cells (n = 12 hiPSC and n = 6 organs) were normalized separately.

**Bioinformatic analyses**. The R package ggplot2 2.0.0 was used for plotting[65].

*Gene expression data:* Gene expression data was downloaded from the Gene Expression Omnibus (GEO) database (GSE66302)[7] and was normalized with the R package edgeR 3.2.4[7, 66, 67]. Briefly, after applying a cutoff value of four reads, the data was normalized with the weighted trimmed mean of M values (TMM) method. The multi-dimensional scaling plots were also generated with R package edgeR 3.2.4.

*DNA methylation data:* A genic and CGI-centric annotation was used for the 450 K CpG probes[8, 35]. Briefly, the genome was divided into five regions for the genic annotation: the intergenic region (> 10 kb from the nearest TSS), the distal promoter (−10 to 1.5 kb from the nearest TSS), the proximal promoter (−1.5 kb to

+ 500 bp from the nearest TSS), the gene body ( + 500 bp to 3′ end of the gene) and the downstream region (3′ end to + 5 kb from 3′ end). For the CGI-centric annotation, the genomic locations of CpG islands were downloaded from the UCSC browser[68], and the CpGs were assigned as non-CGI, CGI, CGI shore or CGI shelf. For both, multidimensional scaling and clustering, Euclidian distance was used. For the clustering, average linkage was applied.

*Differential expression and correlation with methylation:* The differentially expressed genes between W8-12 and W20-22 (with W16-18 in between), were identified with R package edgeR 3.2.4 using a FDR < 0.01[66, 67]. For the organs that only had one replicate at one or more time points, the mean of all biological coefficients of variation (BCVs) of the organs with at least two replicates at all three time points was used as dispersion value. The CpGs in the nearest proximal promoters and gene bodies of all uniquely differentially expressed genes per organ were selected and the difference in beta value between W8-12 and W20-22 was calculated.

*Hypermethylation and hypomethylation:* Relative hypermethylated and hypomethylated CpGs were defined as pairwise difference of > 0.2 or < 0.2 in beta values, respectively, in the sample of interest compared to the other samples.

*Chromatin state segmentations:* Chromatin state segmentations for the available organs (fetal adrenal, amnion, fetal heart atrium and ventricle, fetal intestine, fetal kidney, adult liver, fetal lung, fetal muscle, adult pancreas, placenta, adult fibroblasts (skin), fetal stomach) were downloaded from the Epigenomics Roadmap Project[42]. The enrichment was calculated as odds ratio (OR).

*Differentially methylated regions:* Organ-specific DMRs were identified as previously described[8, 35]. Briefly, DMRs were defined by three consecutive CpGs sharing a common feature (hyper- or hypomethylation) with a maximum of 1 kb between them and not more than three CpGs that did not have the common feature.

**External data**. Gene expression and DNA methylation data were downloaded from NCBI's Gene Expression Omnibus (GSE66302[7]; GSE56515[8]; GSE30654[5]; GSE61461[46]).

**Data availability**. All data are available and have been deposited in the NCBI's Gene Expression Omnibus under the accession number GSE76641.

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

## Acknowledgements

We would like to acknowledge H. Locher for the help with tissue dissection, the Centre for Contraception, Abortion and Sexuality (CASA) in Leiden and The Hague for the collection of the human fetal material, C. Freund for advice and support regarding reprogramming, V. Schwach for providing RNA from hESC-NKX2.5[eGFP/w], R. Ramakrishnan for testing primers and W. Arindrarto for technical support on the bioinformatics analyses. S.C.d.S.L. is supported by the Netherlands Organisation for Scientific Research (NWO, ASPASIA 015.007.037) and the Interuniversity Attraction Poles (IAP, P7/07); M.S.R. and C.L.M. are supported by the Bontius Stichting; F.C. and E.J.P.d.K. by the Bontius Stichting, Stichting DON, the Dutch Diabetes Research Foundation and JDRF; B.T.H. and R.C.S. by the European Union's Seventh Framework Program IDEAL (FP7/2007-2011) under grant agreement No. 259679.

## Author contributions

M.S.R., S.M.C.d.S.L., C.L.M., F.C. and E.J.P.d.K. conceived and designed the experiments. M.S.R., R.C.S., L.v.I., M.M.G.F., N.H., M.B., H.E.D.S. and K.S. performed the

experiments. M.S.R., R.C.S., M.B., C.L.M., K.S., B.T.H. and S.M.C.d.S.L. analysed the data. M.S.R., C.L.M. and S.M.C.d.S.L. wrote the manuscript. All authors read and approved the manuscript.

## Additional information

**Competing interests:** The authors declare no competing financial interests.

