## [Peer Review File · Nature Communications]

Reviewer #1 (Remarks to the Author):

Roost et al have put together and analysed a large collection of transcriptional and DNA methylation data from 21 tissues during fetal development. They compare and correlate the two data types, aiming to gain insights into human cell lineage specification and organogenesis. Using different isogenic cell types, they provide evidence of epigenetic memory upon iPSC generation from fetal brain.

This work is an expansion of the lab's previous DNA methylation data from 4 human tissues (Sliker et al 2015 PLoS Genetics). This is clearly a very useful resource for the scientific community, as these are difficult tissues to source. Overall, the analysis is presented in a very simple, straightforward way, which is a refreshing change from the often over-intricate analyses done on epigenomic data. The resource aspect aside, the novelty of the paper heavily lies on the comparison of isogenic tissues and their use in iPSC generation. This is a strong point, but I felt the authors didn't make full use of this unique aspect. To detail this and other queries:

1. Seems like there was a missed opportunity to also reprogram non-isogenic cells from the same tissue to weigh the effects of genetics vs epigenetics on iPSC generation. As the authors state in the introduction: "The main contribution to the variation between hiPSCs and hESCs has also been suggested to be the genetic background instead of epigenetic memory. However, data to distinguish between these two possibilities is currently lacking." I don't think this question was fully answered by the authors. Although they show that epigenetic memory alone can differentiate iPSCs, a genetic component may also be at play to some (or higher) extent.

2. A similar question can be asked about tissue specification. That is, what drives larger DNA methylation differences: tissue type or genetics? Put differently, is DNA methylation more powerful at predicting tissue type or individuals? One may expect, for example, that early in development the larger differences are genetically driven. The authors should be able to make some cross-individual comparisons to get at least a glimpse of the answer to this question.

3. A number of SAGE libraries have well under 10M reads (Supp. Fig. 1a), which sounds short. Can the authors comment on the detectability limits for this sequencing depth and how it affects the power for identifying DE genes?

4. I find the highlighting of clusters in Fig 1b misleading. Whilst it is stated in the main text that the clusters are for liver (blue) and brain/spinal cord (green), the figure is suggestive of endoderm and ectoderm clusters. The latter don't make tight clusters, being more spread out and intermingled

with mesoderm-derived tissues. As such, I find the statement “the organs primarily separated on the basis of their germ layer of origin (extraembryonic, ectoderm, mesoderm, endoderm)” over-simplistic. The authors should provide a more accurate description of what is seen in the figure.

5. I wonder if looking at single tissue-specific changes is too restrictive. By that I mean expression and methylation events that are seen in only one tissue, which the authors focused on. With such a large array of tissues, it is unsurprising that these become hard to find. This seems to severely limit the statistical power at points – e.g., when analyzing the relationship between methylation and DEGs (especially down-DEGs). An argument could be made to also look at genes with high (but not absolute) tissue specificity, i.e., expressed in a small subset of tissues.

6. I find the section title “Organ-specific transcriptional changes correlate with changes in DNA methylation during human fetal development ” another over-simplification that is also a skewed view of the data in Fig. 2d. Clearly the most common pattern is one of non-existent or very mild correlations. Additionally, the increase in methylation at down-DEGs of eye, liver and muscle are based on only 10, 3 and 2 genes, respectively. This is an extremely limited subset to be extrapolating genome- and organism-wide conclusions from. The functionally relevant examples that the authors provide are interesting and potentially causally linked, but the bottom line is that the overall DNA methylation pattern does not predict transcriptional output. Again, all this requires is accurate description and interpretation of the data.

7. The data supporting skewed neural differentiation of brain-derived iPSCs is weak (Supp. Fig. 4g). Firstly, the SOX9 staining is far from clear. Secondly, the authors should provide numbers of TUBB3+ and SOX9+ cells, with the appropriate statistics. Thirdly, these data could be complemented by RT-qPCR data on a larger set of markers.

8. In Slieker et al there was a strong point made about dynamic methylation at enhancers. Why aren't they analysed here?

9. In Fig. 3b,d I would suggest excluding placenta from the heatmaps, as it may otherwise mislead the reader into thinking that there are no specific methylation changes in this tissue.

Reviewer #2 (Remarks to the Author):

This manuscript is entitled “DNA methylation and transcriptional trajectories during human development and reprogramming of isogenic induced pluripotent stem cells (iPSCs).” The manuscript reports a transcriptional and DNA methylation “atlas” that covers 21 organs during fetal development and includes details of transcripts, hyper- and hypo-methylated regions of the genome at the whole-organ level. The authors then derive iPSCs and note the retention of some methylation marks that might predispose differentiation to specific lineages.

The paper is well-written and the data are of interest to a broad community. Nonetheless, it is a bit confusing as the manuscript reports and uses the same data as previously reported (in two papers in 2015 (references 7 and 8)). One of the references reports transcriptional profiles and the other reports DNA methylation. So there is little new in the first half of the paper. The new experiments in this manuscript are those that include derivation of iPSC lines and maintenance of some of the brain-specific methylation patterns. This is not a particularly novel observation either. Thus overall this manuscript is sound but is not novel.

LEIDEN UNIVERSITY MEDICAL CENTER

Dear Editor,

Please find enclosed our revised manuscript entitled “**DNA methylation and transcriptional trajectories during human development and reprogramming of isogenic induced pluripotent stem cells**” (NCOMMS-16-28633).

We appreciated the comments of the reviewers as they were very helpful in improving the quality and clarity of our manuscript. We have modified the manuscript accordingly and provide a marked-up version of the manuscript for ease of consideration (relevant changes highlighted in blue). We have also adapted the Figures 1 and 4 as the reviewers suggested. Moreover, we performed further analysis using chromatin state segmentations from the Epigenomics Roadmap Project and assessed the neural differentiation by quantitative PCR. These results have been included in the Supplementary Figures 3 and 4 and in the manuscript.

We appreciate your time and look forward to your response.

Sincerely,
Susana M. Chuva de Sousa Lopes
On behalf of all co-authors

Reviewer #1 (Remarks to the Author):

The authors have addressed my questions/suggestions appropriately. The manuscript is now more balanced regarding data interpretation and the data on iPSC epigenetic memory improved.

Only one minor comment on the description of the TUBB3/SOX9 results: the text still refers to numbers of positive cells ("However, the brain-hiPSC culture contained more TUBB3-positive neural derivatives and less SOX9-positive neural-progenitor cells than the skin-hiPSC culture"), whereas now it is RT-qPCR data that is displayed, so the text should refer to higher or lower levels of expression.

Reviewer #2 (Remarks to the Author):

The manuscript by Roost et al entitled "DNA methylation and transcriptional trajectories during human development and reprogramming of isogenic induced pluripotent stem cells." The authors report here on the methylation and transcription in tissues and induced pluripotent stem cells derived from 21 organs.

The authors address the concern that the data has largely been published elsewhere with an explanation of the new material. In addition, they have addressed the novelty, or lack thereof, as well.

The manuscript remains a resource in terms of analyses on tissues that are difficult to obtain. In essence, the value of the manuscript is more in the provision of data than in novel findings.

Reviewer #1 (Remarks to the Author):

The authors have addressed my questions/suggestions appropriately. The manuscript is now more balanced regarding data interpretation and the data on iPSC epigenetic memory improved.

Only one minor comment on the description of the TUBB3/SOX9 results: the text still refers to numbers of positive cells ("However, the brain-hiPSC culture contained more TUBB3-positive neural derivatives and less SOX9-positive neural-progenitor cells than the skin-hiPSC culture"), whereas now it is RT-qPCR data that is displayed, so the text should refer to higher or lower levels of expression.

We thank the reviewer for the positive assessment of our revised manuscript.

We adapted the passage as follows (page 10 line 297-303; Final: Show Markup):

“However, the brain-hiPSC culture showed higher TUBB3 expression and less SOX9 expression than the skin-hiPSC culture, suggesting higher neural differentiation propensity (Supplementary Fig. 4g). Indeed, after 12 days of differentiation, the brain-hiPSC culture showed more TUBB3 expression (Supplementary Fig. 4g) and contained more GFAP-positive neural derivatives (Supplementary Fig. 4h) than the skin-hiPSC culture.”